# Using artificial intelligence-based software for an unbiased discrimination of immune cell subtypes in the fracture hematoma and bone marrow of non-osteoporotic and osteoporotic mice

**Verena Fischer**[1], **Anita Ignatius**[1], **Katharina Schmidt-Bleek**[2], **Georg Duda**[2], **Melanie Haffner-Luntzer**[1,2]*

**1** Institute of Orthopaedic Research and Biomechanics, University Medical Center Ulm, Ulm, Germany,
**2** Julius Wolff Institute, Berlin Institute of Health, Charité, Berlin, Germany

* melanie.haffner-luntzer@uni-ulm.de

## Abstract

It is well established that the early inflammatory response following fracture is essential for initiating subsequent bone regeneration. An imbalance in inflammation, whether within the innate or adaptive immune response, can result in impaired fracture healing. In our previous studies, we demonstrated that, for example, mice with ovariectomy-induced osteoporosis exhibit altered immune cell populations in the early fracture hematoma and bone marrow, leading to delayed healing. These analyses were conducted using conventional FACS/flow cytometry software, where surface marker expression was assessed using a single threshold based on isotype controls—a binary "yes or no" decision. Recent advances have highlighted that immune cell populations are often more heterogeneous, with distinct phenotypic subgroups depending on their polarization status. This has been particularly well documented for macrophage subpopulations (M1, M2, and intermediate polarization states). In light of this, we employed a commercially available artificial intelligence-based clustering software (Cytolution) to more accurately and objectively identify immune cell subpopulations. We re-analyzed flow cytometry raw data from fracture hematoma and bone marrow of non-osteoporotic and osteoporotic mice at day 1 after fracture. Our findings revealed distinct subclusters for granulocytes (27 subclusters), macrophages (7 subclusters), B cells (4 subclusters), and T cells (6 subclusters) within the fracture hematoma and bone marrow. Comparing osteoporotic and non-osteoporotic mice, we observed an increased abundance of a specific B cell subpopulation in osteoporotic mice, alongside a significant reduction of a particular granulocyte subpopulation in the early fracture hematoma. Several subclusters of granulocytes, T cells, and macrophages were also altered in the bone marrow. The specific role of these immune cell subclusters remains to be investigated in the future. These results suggest that AI-based clustering may provide a powerful tool for identifying immune cell

**Data availability statement:** All relevant data are within the manuscript and its Supporting information files.

**Funding:** Collaborative Research Center 1149 "Danger Response, Disturbance Factors and Regenerative Potential after Acute Trauma" (Project-ID 251293561) to AI and MHL; Collaborative Research Center 1444 "Directed Cellular Self-Organisation to Advance Bone Regeneration" (Project-ID427826188) to KSB and GD; the funder did not have any role in the study besides funding.

**Competing interests:** The authors have declared that no competing interests exist.

phenotypes during bone regeneration, offering a more nuanced understanding of flow cytometry data.

## Introduction

Fracture healing is a complex physiological process that involves the coordinated interaction of various cellular and molecular components. Despite extensive research, the intricate mechanisms underlying successful bone repair remain only partially understood. One critical aspect of this process which gained a lot of attention in recent years is the role of the immune system [1–5]. Recent studies have begun to elucidate how immune cells not only participate in the initial inflammatory response following bone injury but also influence later stages of fracture healing. It became evident that many comorbidities, which lead to disturbed fracture healing, are also characterized by a misbalanced early inflammatory response after bone fracture. Well studied examples are comorbidities like polytrauma, ageing and post-menopausal osteoporosis [6–11]. In the latter scenario, we found earlier that post-menopausal, osteoporotic (ovariectomized, OVX) mice display increased presence of neutrophils in the fracture hematoma at day 3 after fracture, while no differences in immune cell numbers were found at day 1 after fracture [12]. In the bone marrow, OVX mice displayed increased numbers of B lymphocytes, while neutrophils and T lymphocytes were significantly reduced, already at 1 day after fracture. Furthermore, we detected a more pronounced pro-inflammatory cytokine storm in OVX mice after fracture [12]. At later healing stages, OVX mice displayed delayed bone formation and disturbed fracture healing outcome, which was shown to be partially due to disturbed immune cell-bone cell interactions [8,12,13].

In general, upon the occurrence of a fracture, the immune system is rapidly activated, leading to an inflammatory response that serves to clear debris and recruit essential cells to the injury site [14]. This initial phase, characterized by the infiltration of neutrophils, macrophages, mast cells and lymphocytes, sets the stage for subsequent healing phases. Neutrophils are regarded as the first line of defence leading to debris clearance and a robust proinflammatory response. Macrophages, in particular, exhibit remarkable plasticity, transitioning from a pro-inflammatory (M1) phenotype in the early stages to an anti-inflammatory and tissue reparative (M2) phenotype as healing progresses [10,14]. This dynamic shift is crucial for the resolution of inflammation and the promotion of tissue regeneration. Additionally, T and B lymphocytes have been shown to heavily influence bone repair [2,15,16]. Specific T cell subsets like Th17or TEMRA cells contribute to the regulation of osteoclast and osteoblast activity through the secretion of cytokines, thus impacting bone resorption and formation [16,17].

In numerous studies, immune cell populations in the fracture hematoma and other tissues are commonly characterized using flow cytometry. The conventional approach to analyzing such data, as employed in our previous studies [12,18], involves identifying cell types based on the expression of stained surface markers,

using a single threshold often determined by isotype controls. This method yields a binary "positive" or "negative" outcome for each marker. Alternatively, manual gating can be used to distinguish subpopulations based on "low," "intermediate," or "high" expression of specific surface markers [19]. However, the placement of these cut-offs can be challenging and may introduce bias, influenced by the investigator's interpretation of the data. To address this limitation, we explored the application of a commercially available artificial intelligence- and machine learning-based clustering software ("Cytolution") to perform a more objective and unbiased analysis of immune cell subpopulations. Using this software, we re-analyzed FACS source files from our earlier work [12], focusing on immune cell subpopulations in the early fracture hematoma and bone marrow of osteoporotic (OVX) and non-osteoporotic (sham-OVX) mice. This paper outlines the process and highlights the distinct subpopulations identified through this advanced approach.

## Materials and methods

### Animals

The source data used for that study was generated earlier [12]. 12 female C57BL/6J mice were used for that study under the approval of the local ethical committee (No. 1079 and 1184, Regierungspräsidium Tübingen). All animal experiments were in compliance with international regulations for the care and use of laboratory. Animals were housed in groups of two to four animals per cage ($370\,cm^2$) on a 14-h light and 10-h dark circadian rhythm with water and food ad libitum. Mice aged 3–4 months underwent bilateral sham operation (n=6) or OVX (n=6) as described previously [12]. Osteotomy was performed 8 weeks after sham/OVX, at that time point OVX mice already displayed postmenopausal osteoporosis. Briefly, the osteotomy was created at the femur diaphysis using a 0.4 mm Gigli wire saw (RISystem, Davos, Switzerland) and stabilized by a semi-rigid external fixator (RISystem). Animals were euthanized at day 1 after fracture surgery using an isoflurane overdose and cardiac blood withdrawal. The fracture hematoma was harvested from the right fractured femur, while the bone marrow was harvested from the intact left femur [12].

### Flow cytometry procedure

As described earlier [12], the contralateral bone marrow was flushed out using phosphate-buffered saline and passed through a 70-µm cell strainer (Corning Inc., Durham, NC, USA). The fracture hematoma was minced and also passed through a 70-µm cell strainer to obtain single-cell suspension. Both hematoma and bone marrow were subjected to erythrolysis to remove erythrocytes from the samples. FACS analysis was run on an LSR II flow cytometer (BD Bioscience) with the following antibodies: CD3e (PE-Cy7, Affymetrics eBioscience, #25–0031, 1:100); CD4 (APC-e780, Affymetrics eBioscience, #47–0041, 1:200); CD8a (APC, Affymetrics eBioscience, #17–0081, 1:800); CD11b (AlexaFlour700, Affymetrics eBioscience, #56–0112, 1:400); CD19 (PE, Affymetrics eBioscience, #12–0193, 1:400); F4/80 (FITC, Affymetrics eBioscience, #11–4701, 1:50); Ly6G (V450, BD Bioscience, #560603, 1:400). 7AAD dye was used to discriminate living from dead cells. FCS files were exported from the flow cytometer.

### Flow cytometry analysis

FCS files were imported into the Cytolution software (Cytolytics GmbH, Tübingen, Germany). In the first step, the software checked if the used marker panels and fluorophores matched between all samples. Then, the compensation data were read out from the.fcs files and displayed in dot plots (see Fig 1 (2) for examples). At this step, the user can manually check again the compensation and adjust if needed. No manual adjustments were made for that dataset. The next step is additional data quality control, which includes the automatic control of the time channel and removal of cell debris and cell aggregates. In this step, the user is able to manually select live/dead cells based on a viability dye (see Fig 1 (2)). This was done for that dataset based on 7AAD staining (highly positive cells were excluded as dead cells from further analysis). The third step is data transformation. In this step, the program automatically determines

## Flow Cytometry Experiment - Workflow

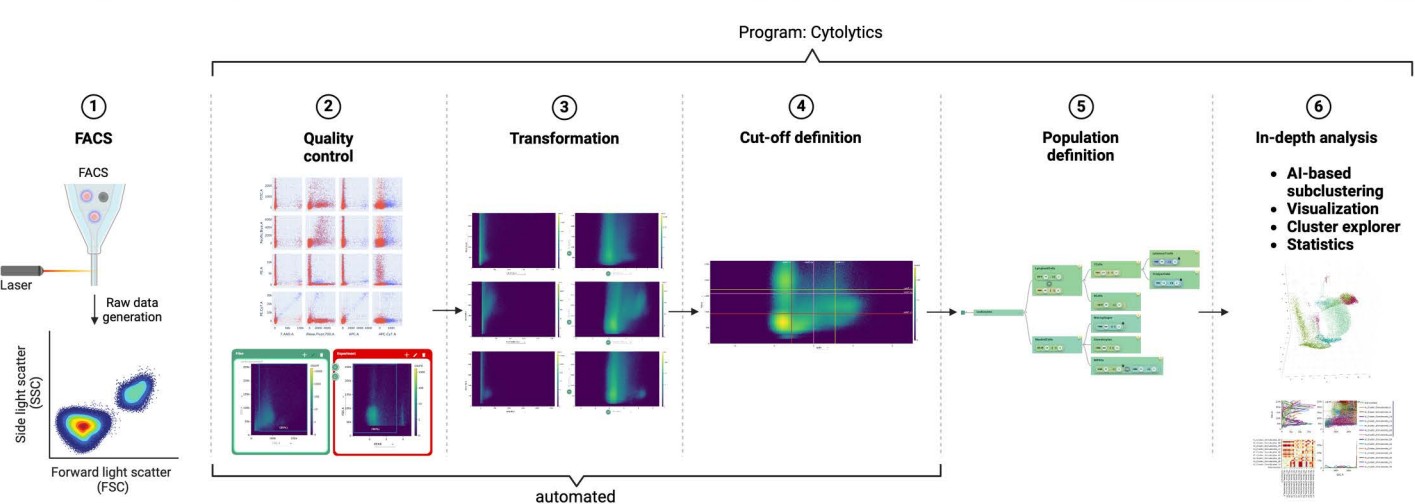

**Fig 1. Workflow of the flow cytometry experiment and the software-based analysis.** (1) Raw data were generated by flow cytometry. (2) FCS files were imported into the Cytolution software. Initial quality control included panel check, compensation data check, automatic control of the time channel and removal of cell debris and cell aggregates. In this step, the user is able to manually select live/dead cells based on a viability dye. (3) The third step is data transformation. In this step, the program automatically determines the needed transformation parameters for each dye dependent on the input data. Optionally, the user is able to manually do the transformation, which was not performed for the present dataset. (4) The next steps are cutoff definition and population definition. The program automatically determines multiple cutoff values per dye based on the input data (+/-, low, +/+). Optionally, the user is able to manually shift the cutoffs, which was not performed for the present dataset to obtain an unbiased view on the data. (5) Based on these cutoffs, a population tree can be built by the user. In the present study, we determined the populations CD11b+Ly6G+F4/80- granulocytes, CD11b+Ly6G-F4/80+ macrophages, CD11b+Ly6G+F4/80+ MDSCs, CD19+ B-cells, CD3+ T-cells, CD3+CD8+ cytotoxic T-cells, CD3+CD4+ Thelper-cells as of interest for us. (6) The program calculates multiple subclusters based on AI/machine learning-algorithms. The subclusters can either be independent from the defined population tree or they are classified as subclusters within the pre-defined subpopulations from the population tree. All pre-defined populations and AI-generated subclusters can be visualized in 2D or 3D UMAPs (or t-SNE) based on their expression patterns. Furthermore, absolute cell numbers or ratios can be exported or displayed as graphs in the program itself. The most innovative part is to further explore the AI-based subclusters in the Cluster explorer. In this part of the program, the user is able to see why distinct subclusters were generated based on SSC, FSC and other marker expressions.

the needed transformation parameters for each dye dependent on the input data (see Fig 1 (3) for examples). Optionally, the user is able to manually do the transformation, which was not performed for the present dataset due to the accuracy of the automatic transformation. The next steps are cutoff definition and population definition. The program automatically determines multiple cutoff values per dye based on the input data (+/-, low, +/+) (see Fig 1 (4) for example). Optionally, the user is able to manually shift the cutoffs, which was not performed for the present dataset to obtain an unbiased view on the data. Based on these cutoffs, a population tree can be built by the user. This is the first step where the user has to manually select categories and cut-off values to annotate specific cell-populations. However, AI-based population clustering is in general independent from the population tree. In the present study, we determined the populations CD11b+Ly6G+F4/80- granulocytes, CD11b+Ly6G-F4/80+ macrophages, CD11b+Ly6G+F4/80+ MDSCs, CD19+ B-cells, CD3+ T-cells, CD3+CD8+ cytotoxic T-cells, CD3+CD4+ Thelper-cells as of interest for us (Fig 1 (5)). Additionally, the program calculates multiple subclusters based on AI/machine learning-algorithms (described in more detail in the next section). The subclusters can either be independent from the defined population tree or they are classified as subclusters within the pre-defined subpopulations from the population tree. All pre-defined populations and AI-generated subclusters can be visualized in 2D or 3D dimensionality reduced maps (UMAP, Tsne, Cyto) based on their expression patterns. Furthermore, absolute cell numbers or ratios can be exported or displayed as graphs in the

program itself. For the present study, we exported the ratio of the populations/clusters based on living cells and finished the analysis by GraphPad prism. The most innovative part is to further explore the AI-based subclusters in the Cluster explorer. In this part of the program, the user is able to see why distinct subclusters were generated based on SSC, FSC and all other marker expressions. Therefore, we were able to further investigate the different subclusters which were of interest for us.

### AI-based algorithm

Cluster identification is performed using the "PARC" algorithm, an unsupervised learning and open-source algorithm. PARC autonomously detects clusters and is therefore making the distinctions solely based on the data. Once set and defined, the "PARC" algorithm operates independently of user-defined population specifications, ensuring that cluster formation and composition remain unaffected by user decisions (e.g., like population definition criteria). The user is only responsible for defining how the clusters are annotated, determining which subtypes receive specific labels. However, the clusters themselves are computed beforehand and are not influenced by this annotation process. The Cluster Explorer utilizes the median fluorescence intensity (MFI) of each selected cluster (subpopulation) for comparison, highlighting the most significant differences in expression between clusters. It identifies the largest distances between clusters, revealing where the greatest variations in expression occur for a given channel in the compared clusters. To verify the stability of the algorithm, we ran the same experiment in duplicates using the subsampling setting while reducing the amount of input data for the algorithms. This approach helps determine how stable the results are and whether there is a threshold at which clusters or populations become undetectable due to excessive downsampling. A recommended strategy is to downsample to one-third of the original data and allow the pipeline to recalculate.

### Statistics

Data are expressed as mean ± standard deviation. Comparisons between sham-OVX and OVX were performed using two-tailed Student's t-test. Values of $p<0.05$ were considered to be statistically significant and are displayed as exact numbers. Statistical analysis was performed using Graph Pad Prism 9.0 software (GraphPad Software, La Jolla, CA, USA).

## Results

### Immune cell population analysis based on pre-defined population tree

In a first step, we used the automatically generated cut-off values for our stained surface markers CD11b, Ly6G, F4/80, CD3, CD4, CD8 and CD19 to evaluate the following user pre-defined immune cell populations: CD11b+Ly6G+F4/80- granulocytes, CD11b+Ly6G-F4/80+ macrophages, CD11b+Ly6G+F4/80+ MDSCs, CD19+ B-cells, CD3+ T-cells, CD3+CD8+ cytotoxic T-cells, CD3+CD4+ T helper-cells. For the fracture hematoma, we did not detect differences in the numbers of the immune cell populations between sham-OVX and OVX mice (Fig 2A–G). A 3D UMAP of all selected cell populations for all samples is displayed in Fig 2H. Fig 2I displays a 3D UMAP of all selected cell populations in the samples from sham-OVX mice only, whereas OVX mice samples are displayed in Fig 2J. These UMAPs show similarities between the selected populations after a dimensionality reduction.

For the bone marrow, we did not detect differences in macrophage, MDSC and B-cell ratio between sham-OVX and OVX mice, while granulocyte numbers as well as general T-cell numbers were significantly reduced in OVX mice. In contrast, the T-cell subpopulations cytotoxic T-cells and T helper-cells were significantly increased (Fig 3A–G). A 3D UMAP of all selected cell populations for all samples is displayed in Fig 3H. Fig 3 I displays a 3D UMAP of all selected cell populations in the bone marrow samples from sham-OVX mice only, whereas OVX mice samples are displayed in Fig 3J. The UMAPs displayed in Fig 2I, J and Fig 3I, J indicated that there might differences between sham-OVX and OVX mice specially in the granulocyte and B-cell population. This was explored further in the next step.

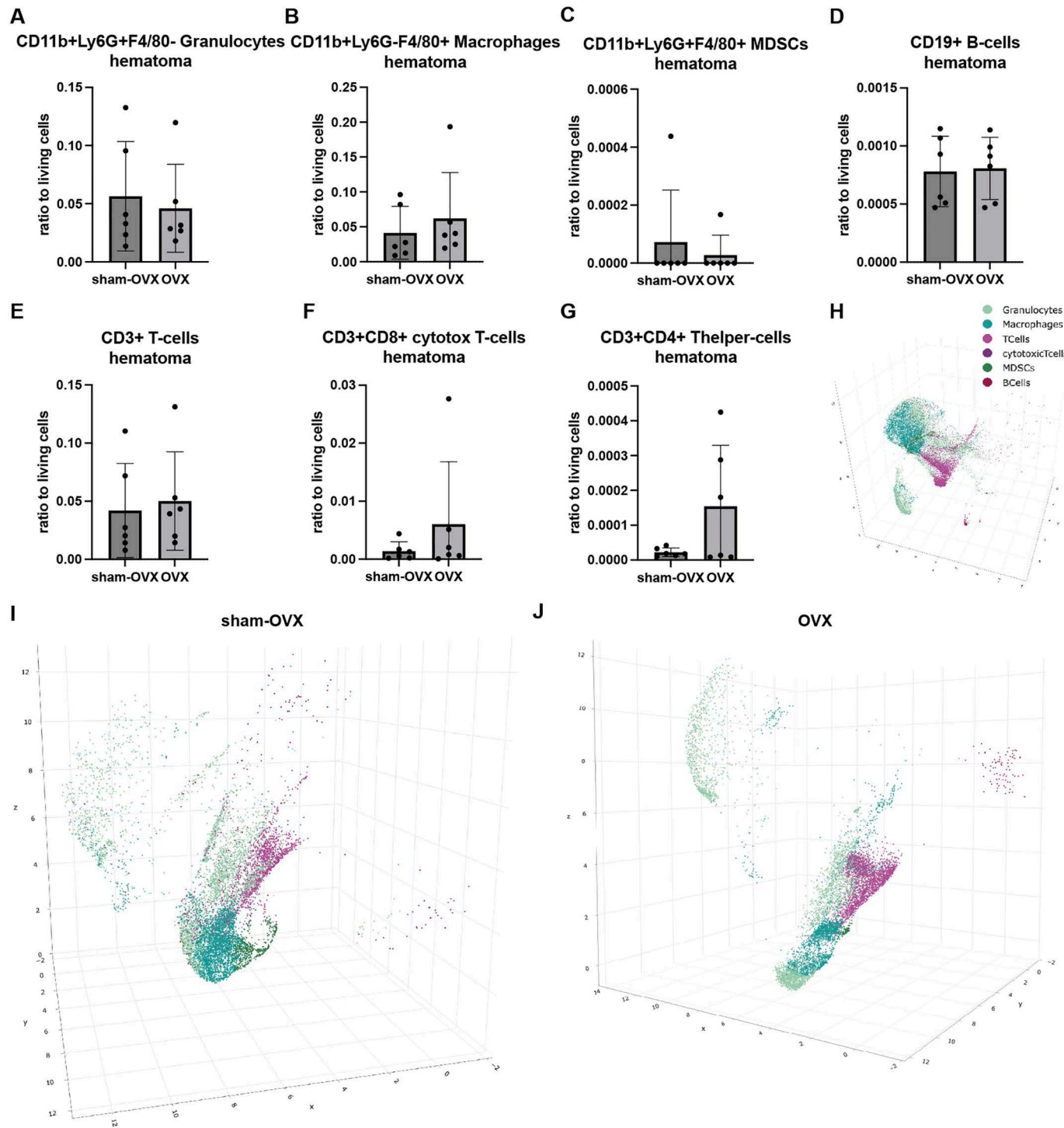

**Fig 2. Immune cell populations in the fracture hematoma.** We used the automatically generated cut-off values for our stained surface markers CD11b, Ly6G, F4/80, CD3, CD4, CD8 and CD19 to evaluate the following user pre-defined immune cell populations: A) CD11b+Ly6G+F4/80- granulo-cytes, B) CD11b+Ly6G-F4/80+ macrophages, C) CD11b+Ly6G+F4/80+ MDSCs, D) CD19+ B-cells, E) CD3+ T-cells, F) CD3+CD8+ cytotoxic T-cells, G) CD3+CD4+ Thelper-cells. H) A 3D UMAP of all selected cell populations for all samples is displayed. I) 3D UMAP of all selected cell populations in the hematoma samples from sham-OVX mice. J) 3D UMAP of all selected cell populations in the hematoma samples from OVX mice.

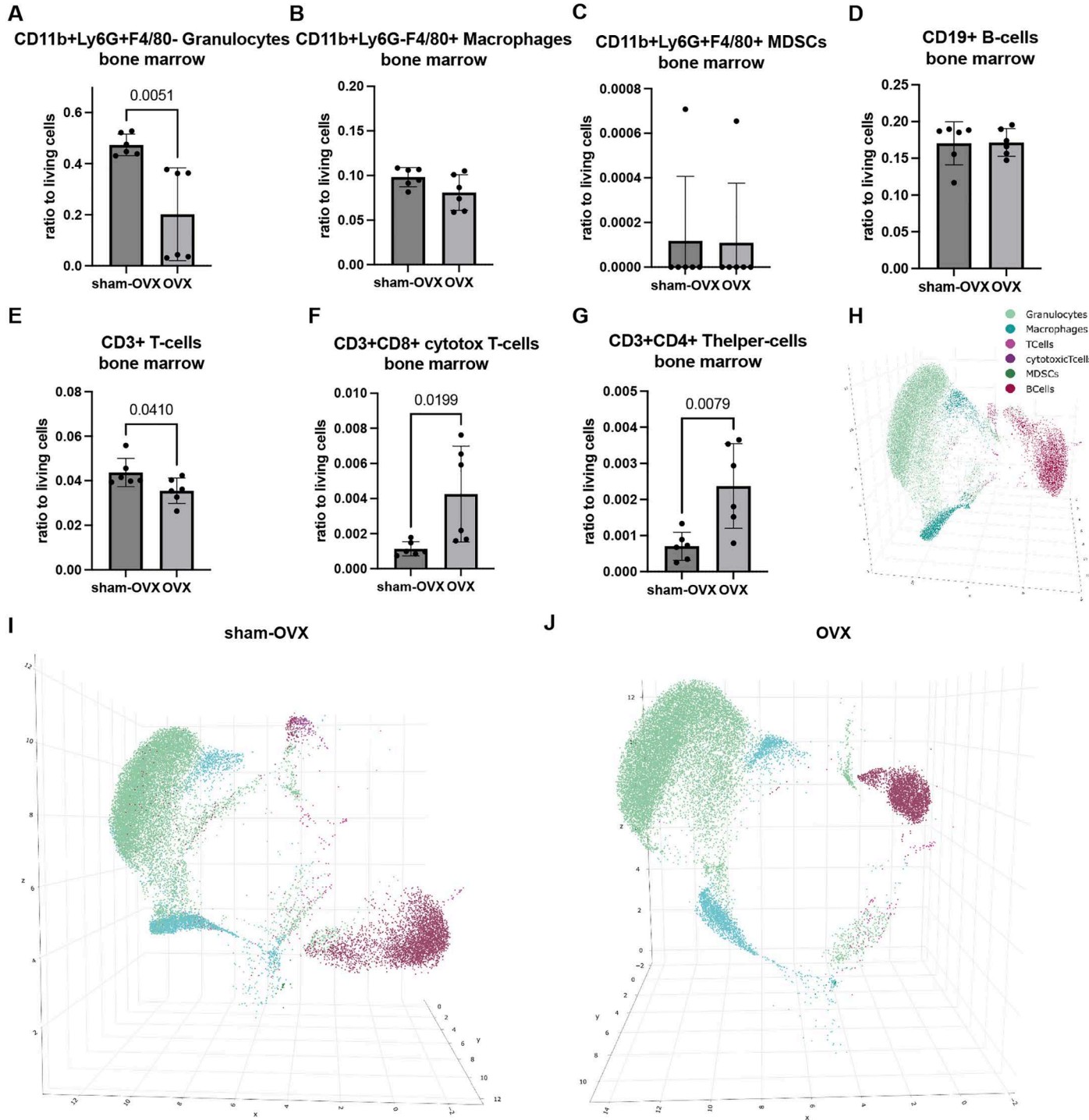

**Fig 3. Immune cell populations in the bone marrow.** We used the automatically generated cut-off values for our stained surface markers CD11b, Ly6G, F4/80, CD3, CD4, CD8 and CD19 to evaluate the following user pre-defined immune cell populations: A) CD11b+Ly6G+F4/80- granulocytes, B) CD11b+Ly6G-F4/80+ macrophages, C) CD11b+Ly6G+F4/80+ MDSCs, D) CD19+ B-cells, E) CD3+ T-cells, F) CD3+CD8+ cytotoxic T-cells, G) CD3+CD4+ Thelper-cells. H) A 3D UMAP of all selected cell populations for all samples is displayed. I) 3D UMAP of all selected cell populations in the bone marrow samples from sham-OVX mice. J) 3D UMAP of all selected cell populations in the bone marrow samples from OVX mice.

## Immune cell subpopulation analysis based on AI-based subclusters

Next, we had a look at the AI-based subclusters which were generated automatically by the program. This analysis was based on input data from all.fcs files in this experiment. The program determined 3 subclusters for T-cells, 2 subclusters for cytotoxic T-cells, 1 subcluster for T helper-cells, 4 subclusters for B-cells, 1 subcluster for MDSCs, 7 subclusters for macrophages and 27 subclusters for granulocytes. All cell subpopulations are displayed in the 3D UMAP in Fig 4. We also analyzed the ratio of the respective subpopulation towards living cells and if these ratios were different between

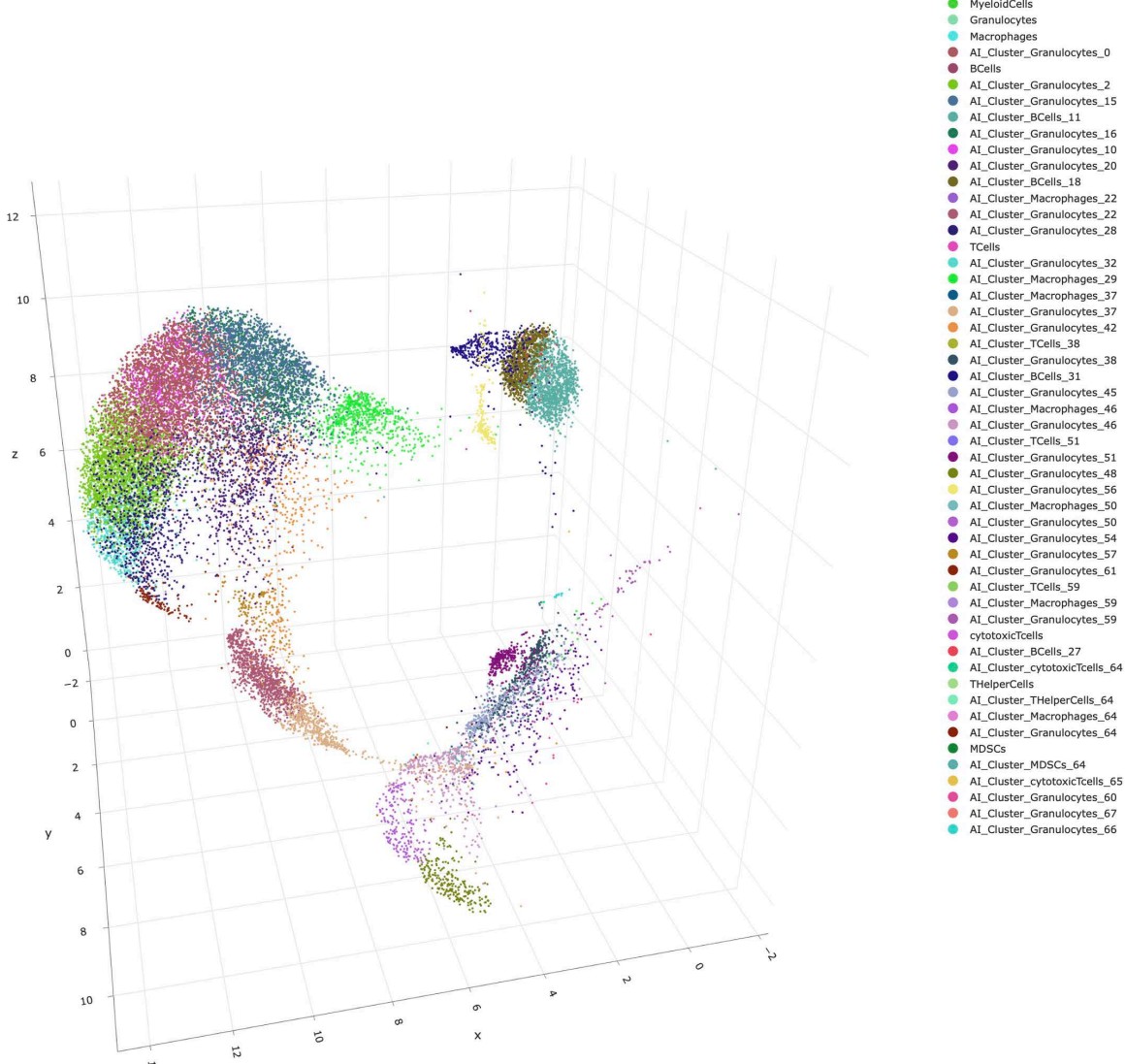

**Fig 4. 3D UMAP of immune cell AI-based subclusters based on all samples of the dataset.** To construct the initial high-dimensional graph, UMAP creates what is known as a "fuzzy simplicial complex." This is essentially a weighted graph, where the weights on the edges represent the probability of two points being connected. To establish these connections, UMAP expands a radius around each point and links points when their radii overlap. The choice of radius is crucial, if it is too small, it results in small, isolated clusters, while a large radius connects everything indiscriminately. UMAP addresses this by selecting a radius locally, based on the distance to each points nearest neighbor. It then introduces "fuzziness" by reducing the probability of connection as the radius increases. Lastly, UMAP ensures that each point is connected to at least its closest neighbor, preserving local structure while balancing it with the global layout. Therefore, a UMAP show similarities between the different clusters after a dimensionality reduction-.

sham-OVX and OVX mice in the hematoma (S1 Table) or the bone marrow (S2 Table). We decided to further explore only the subpopulations which were significantly altered between sham-OVX and OVX mice in the fracture hematoma. We found that one subcluster from the B-cells ("AI_subcluster_Bcells_31") was significantly different between sham-OVX and OVX mice with OVX mice having a higher ratio of this subcluster in the hematoma (Fig 5A). In the bone marrow, this subpopulation did not differ between the groups (Fig 5B). Also, the ratio of one subcluster from the granulocytes ("AI_based_subcluster_granulocytes_0") was significantly reduced in OVX mice both in the hematoma (Fig 5C) and the bone marrow (Fig 5D), with in general low cell numbers in the hematoma. All other subclusters did not differ in the hematoma between sham-OVX mice and OVX mice. In the next step, the subcluster was explored further in the program to see which unique features are displayed by this respective subpopulation. Exemplarily, we show that for the "AI_based_subcluster_granulocytes_0" in this manuscript (Fig 6), as it was different between the two groups in both the hematoma and the bone marrow. We observed that this subcluster of granulocytes display a larger cell size when compared to other granulocyte subclusters and an intermediate granularity (Fig 6A and B). Furthermore, this subcluster displayed a high membrane integrity (as indicated by low 7AAD staining) and an intermediate-to-high Ly6G expression (Fig 6C and D). Also, these cells display an intermediate CD11b expression (Fig 6E and F). These data indicate that these granulocytes are in an intermediate activation status, although of course direct correlations between the stained markers and biological activity cannot be drawn. Further research is needed to characterize this interesting subcluster of granulocytes in more detail. Not only FSC, SSC, DEAD-staining, Ly6G expression and CD11b expression were used by the algorithm to detect subpopulations, but also all other available information. However, since the other markers (CD19, CD3, CD8, CD4, F4/80) are generally not expressed on neutrophils, emphasis was much less from the algorithm on these markers, this is why we do not display them in Fig 6. All subcluster data is displayed in S3 Table.

## Discussion

In this manuscript, we describe the possible use of a software allowing objective AI-based subclustering of immune cell populations based on flow cytometry data. For the presented study, we used previously generated raw data files from fracture hematoma and bone marrow from non-osteoporotic (sham-OVX) and osteoporotic (OVX) mice. These raw data had been previously analyzed by conventional gating strategies based on istotype controls (published in [12]). In the first step, we re-analyzed the same user pre-defined immune cell populations, but this time based on automatically generated cut-off values. We found that in the fracture hematoma, the different immune cell populations did not differ between sham-OVX and OVX mice. This was in line with the data generated previously by conventional gating strategies [12]. In the bone marrow, we found no differences in macrophage, MDSC and B-cell ratio between sham-OVX and OVX mice, while granulocyte numbers as well as general T-cell numbers were significantly reduced in OVX mice. Furthermore, T-cell subpopulations cytotoxic T-cells and Thelper-cells were significantly increased in OVX mice. These data are in line with the previously published results for macrophages, granulocytes, MDSCs, T-cells and T-helper cells. However, we previously detected significantly increased numbers of B-cells in the bone marrow of OVX mice, this was not visible in the re-analyzed data. Also, cytotoxic T-cells were significantly decreased in the previously published data in contrary to the re-analyzed data. Therefore, we can conclude that the chosen data analysis platform and strategy can have a significant impact on the data outcome and because of that, it is always recommended to verify flow cytometry data with another molecular biology method, e.g., immunohistochemical staining or single cell RNA sequencing. Also, the usage of appropriate controls is critical, e.g., isotype staining for easier cut-off definition, which is not implemented yet in the software presented here. However, the biggest advantage of the here described software is that it uses an artificial intelligence/ machine learning-based clustering algorithm to determine subpopulations of immune cells in a more unbiased and objective manner. The subclustering is done based on all available information from the cytometry data set. The software offers several approaches for analyzing immune cell populations. One option is for the investigator to forgo pre-defining immune cell populations based on specific surface marker combinations and instead rely entirely on AI-based clustering. This

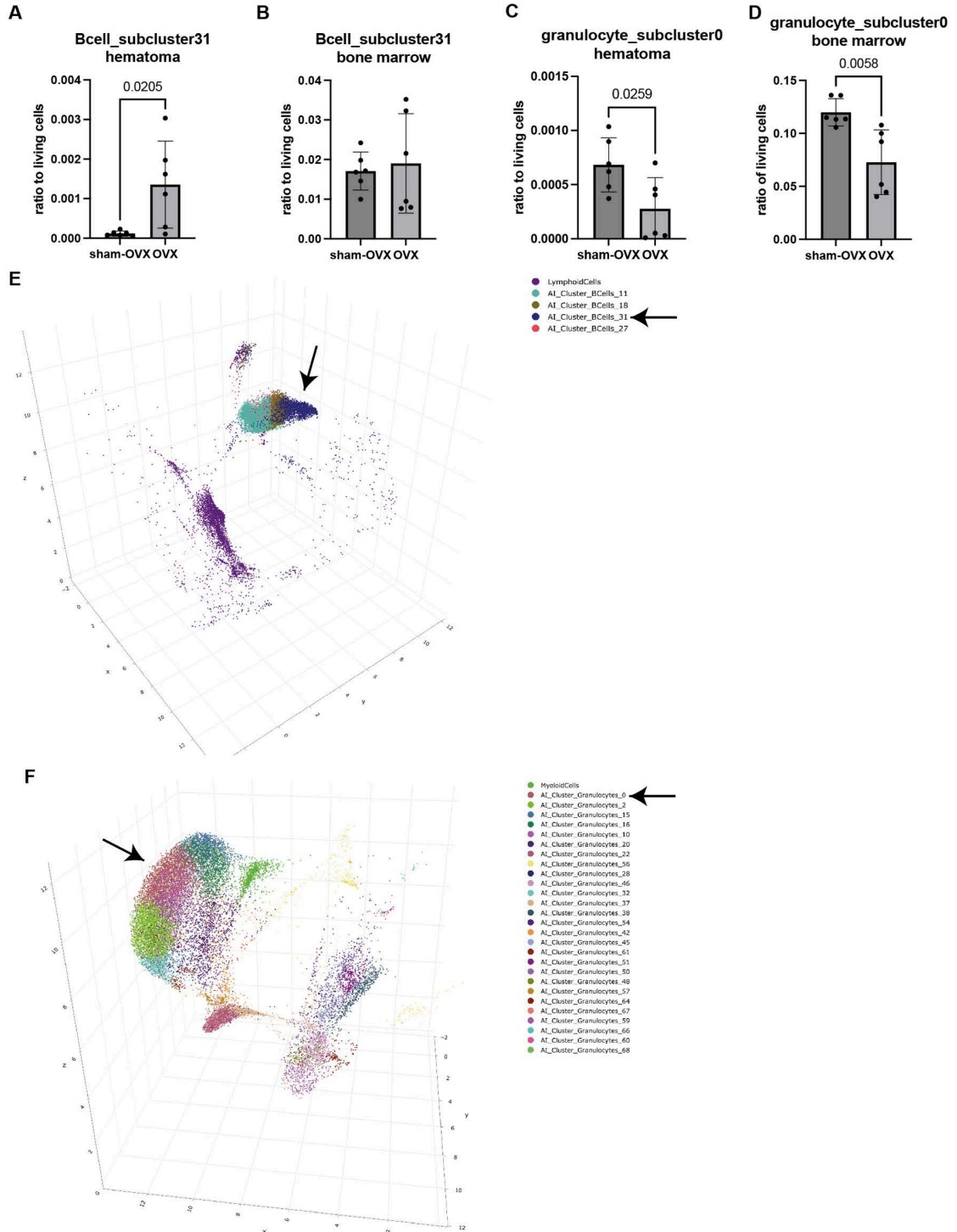

**Fig 5. AI-based subcluster analysis.** Only subclusters which were significantly different between sham-OVX and OVX mice in the hematoma are displayed. Data for all other subclusters are displayed in S1 and S2 Tables. A) "B-cell subcluster31" in the hematoma and B) the bone marrow. C) "Granulocyte_subcluster0" in the hematoma and D) the bone marrow. E) A 3D UMAP of all B-cell subclusters is displayed. F) A 3D UMAP of all granulocyte subclusters is displayed.

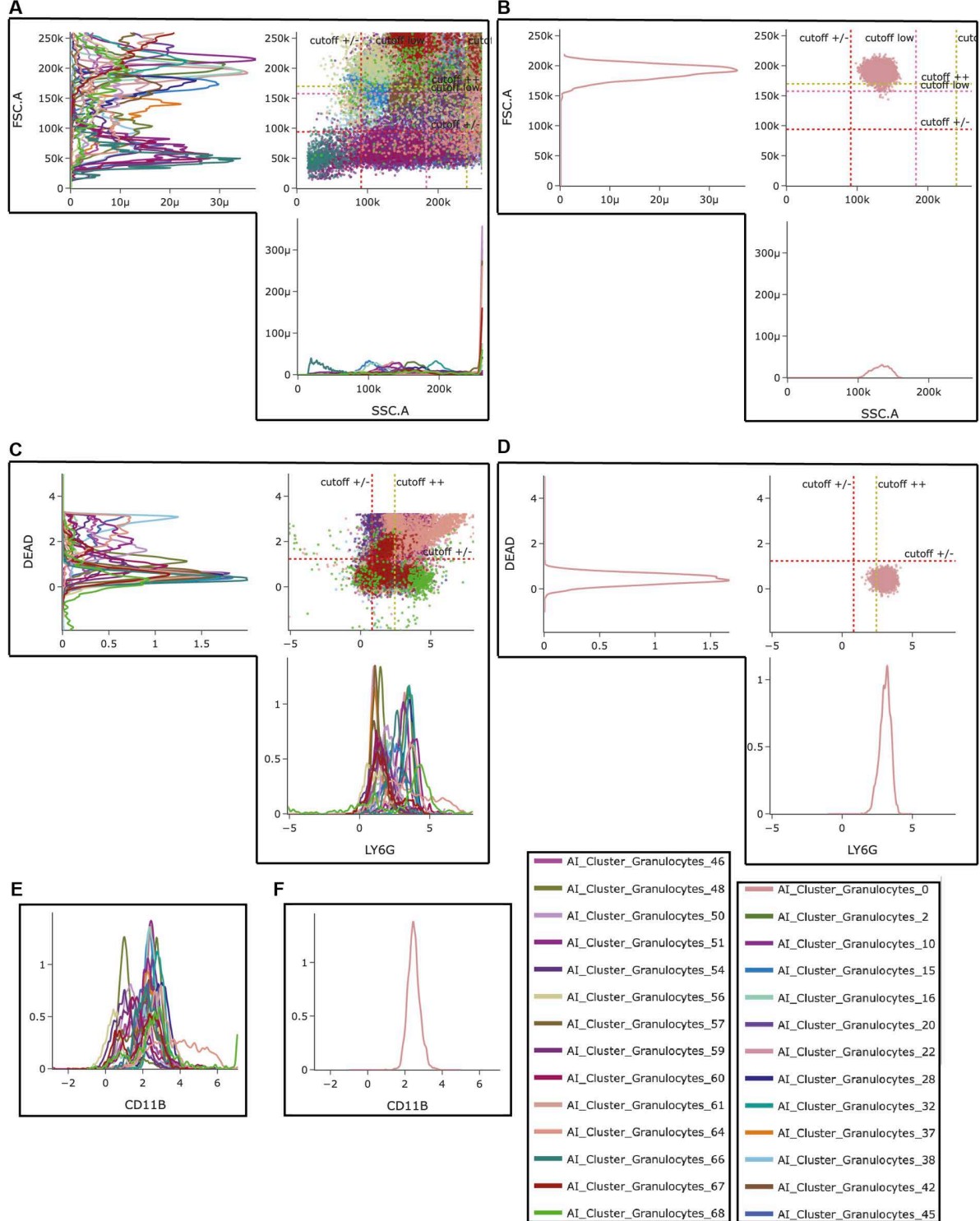

**Fig 6. Cluster explorer analysis for granulocyte subclusters.** We further looked at the unique features of the 27 granulocyte subclusters. A) FCS vs. SSC for all subclusters and B) only for subcluster_0. C) 7AAD (dead staining) vs. Ly6G for all subclusters and D) only for subcluster_0. E) CD11b expression for all subclusters and F) only for subcluster_0. Automatically determined cut-off values can be seen in the graphs.

method is unbiased, as both the cut-off values and cell populations are automatically determined by the software based solely on the input data. In our study, we employed a different approach. We pre-defined specific immune cell populations of particular relevance to fracture healing and then applied the AI-based clustering to further analyze subpopulations within these broader immune cell populations. Notably, the software identified the highest number of subclusters within the granulocytes, which aligns with their characterization in the literature as a highly heterogeneous cell population with multiple activation states [20–22]. We think that the software identifies much more subclusters of granulocytes as for example T lymphocytes, since especially the parameters size, granularity and membrane integrity differ very much in granulocytes during their polarization/differentiation process. Also, we found evidence that neutrophil granulocytes differ between sham-OVX and OVX mice in another previous study [7], which makes them an interesting cell population to look at. And indeed, we found one subcluster of granulocytes which were significantly different in the fracture hematoma and bone marrow of OVX mice at day 1 after fracture. These cells displayed unique features regarding cell size, granularity, membrane integrity and surface marker, indicating an intermediate activation status. However, of course direct correlations between these markers and biological activity cannot be drawn and would need further analysis, e.g., *in vitro* characterization of these cells after sorting. Therefore, it would be recommended for future studies to not only look at cell numbers but also activity of cell populations. Another important aspect is, that these AI-based subpopulations will be even better defined the more markers are included in the flow cytometry panel. Our antibody panel was rather small, it would be important to include more specific markers, e.g., for T-cell subsets, intracellular signaling molecules such as cytokines and growth factors or other metabolites. Therefore, the potential for the usage of AI-based algorithms to detect subpopulations of immune cells in the fracture hematoma would increase with every additionally analyzed marker.

In conclusion, we demonstrated that the usage of an AI-based subclustering strategy is able to gain further information from flow cytometry data to define specific subpopulations of immune cells in the fracture hematoma and bone marrow of mice. This is of great importance for the field of orthopaedic research as it was shown previously that immune cells like macrophages, granulocytes and lymphocytes display specific polarization stages which alter their behavior during the fracture healing process dramatically [16,22–24]. In future studies, we will further analyze specific granulocyte or lymphocyte subpopulations in the fracture hematoma to see how they are affected by postmenopausal osteoporosis and to identify their crosstalk with bone cells. This might help to unravel molecular mechanisms of delayed fracture healing in osteoporotic bone.

## Supporting information

**S1 Table. Subcluster analysis from the hematoma.**
(XLSX)

**S2 Table. Subcluster analysis from the bone marrow.**
(XLSX)

**S3 Table. Subcluster explorer data from all subclusters.**
(XLSX)

## Acknowledgments

We thank Can Pinar from Cytolytics for excellent technical support.

## Author contributions

**Conceptualization:** Verena Fischer, Anita Ignatius, Melanie Haffner-Luntzer.

**Data curation:** Verena Fischer, Melanie Haffner-Luntzer.

**Formal analysis:** Verena Fischer, Melanie Haffner-Luntzer.

**Funding acquisition:** Anita Ignatius, Katharina Schmidt-Bleek, Georg Duda, Melanie Haffner-Luntzer.

**Investigation:** Melanie Haffner-Luntzer.

**Methodology:** Verena Fischer, Melanie Haffner-Luntzer.

**Project administration:** Verena Fischer, Anita Ignatius, Melanie Haffner-Luntzer.

**Supervision:** Katharina Schmidt-Bleek, Georg Duda.

**Visualization:** Melanie Haffner-Luntzer.

**Writing – original draft:** Verena Fischer, Anita Ignatius, Katharina Schmidt-Bleek, Georg Duda, Melanie Haffner-Luntzer.

**Writing – review & editing:** Verena Fischer, Anita Ignatius, Katharina Schmidt-Bleek, Georg Duda, Melanie Haffner-Luntzer.

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
