## [Editor Report · Decision Letter 0]

12 Mar 2025

PONE-D-25-02396Using artificial intelligence-based software for an unbiased discrimination of immune cell subtypes in the fracture hematoma and bone marrow of non-osteoporotic and osteoporotic micePLOS ONE

Dear Dr. Haffner-Luntzer,

Thank you for submitting your manuscript to PLOS ONE. After careful consideration, we feel that it has merit but does not fully meet PLOS ONE’s publication criteria as it currently stands. Therefore, we invite you to submit a revised version of the manuscript that addresses the points raised during the review process.

We look forward to receiving your revised manuscript.

Kind regards,

Mohammed E. Elsalanty

Academic Editor

PLOS ONE

2. Please update your submission to use the PLOS LaTeX template. The template and more information on our requirements for LaTeX submissions can be found at http://journals.plos.org/plosone/s/latex .

Additional Editor Comments:

This study involved a post-hoc analysis of flow cytometry data using an AI algorithm. The aim of the study was to further identify sub-classifications of immune cells involved in the early fracture healing, both at the hematoma and the contralateral bone marrow. The premise of the study was quite intriguing and the findings could provide basis for further investigation of key cellular profiles most important for the critical early stage of bone healing. There were a few areas where the study could improve the impact of its findings:

1. Although the use of AI could reduce human bias inherent in the subjective thresholding, AI algorithms are still marred with subjectivity. Up to this point, there still is significant approximations and guesswork built in. Although it would probably be impossible for the investigators to evaluate how consistent the specific logic was across their samples or, equally important, how reproducible it would be for a future study, it would help if they identify the criteria based on which the algorithm made the distinction between subtypes and the weight given to each.

2. To clarify the point above, the authors identified two subpopulations that were most different: AI_subcluster_Bcells_31 and AI_Based_Subcluster_granulocytes_0 and included some cellular features and expression profiles for each. Was these features determined after the AI determination? or were they the only/main criteria used to make the distinction by the algorithm itself? Were there other criteria?

3. Along the same line, it would be important for future comparisons to include a table where all the identified subpopulations are described in the same manner as the above mentioned two.
---

## [Author Response · Author response to Decision Letter 1]

17 Mar 2025

Point-to-point reply

This study involved a post-hoc analysis of flow cytometry data using an AI algorithm. The aim of the study was to further identify sub-classifications of immune cells involved in the early fracture healing, both at the hematoma and the contralateral bone marrow. The premise of the study was quite intriguing and the findings could provide basis for further investigation of key cellular profiles most important for the critical early stage of bone healing. There were a few areas where the study could improve the impact of its findings:

1. Although the use of AI could reduce human bias inherent in the subjective thresholding, AI algorithms are still marred with subjectivity. Up to this point, there still is significant approximations and guesswork built in. Although it would probably be impossible for the investigators to evaluate how consistent the specific logic was across their samples or, equally important, how reproducible it would be for a future study, it would help if they identify the criteria based on which the algorithm made the distinction between subtypes and the weight given to each.

Answer: We thank the editor for this important comment. Indeed, verifying the consistency of an AI-based algorithm is very important. The best way to test for stability is to run the same experiment in duplicates using the subsampling setting while reducing the amount of input data for the algorithms. This approach helps determine how stable the results are and whether there is a threshold at which clusters or populations become undetectable due to excessive downsampling. A recommended strategy is to downsample to one-third of the original data and allow the pipeline to recalculate. Cluster identification is performed using the "PARC" algorithm, an unsupervised learning and open-source algorithm. PARC autonomously detects clusters and is therefore making the distinctions based on the data (see also explanation to the second point). We indeed verified our data by this approach and have now added this information to the manuscript on page 7.

2. To clarify the point above, the authors identified two subpopulations that were most different: AI_subcluster_Bcells_31 and AI_Based_Subcluster_granulocytes_0 and included some cellular features and expression profiles for each. Was these features determined after the AI determination? or were they the only/main criteria used to make the distinction by the algorithm itself? Were there other criteria?

Answer: Supopulation identification is solely performed using the "PARC" algorithm, an unsupervised learning and open-source algorithm. PARC autonomously detects clusters/supopulations and is therefore making the distinctions based on the data only. Once set and defined, the "PARC" algorithm operates independently of user-defined population specifications, ensuring that cluster formation and composition remain unaffected by user decisions (e.g. like population definition criteria). The user is only responsible for defining how the clusters are annotated, determining which subtypes receive specific labels. However, the clusters themselves are computed beforehand and are not influenced by this annotation process. The Cluster Explorer utilizes the median fluorescence intensity (MFI) of each selected cluster (subpopulation) for comparison, highlighting the most significant differences in expression between clusters. It identifies the largest distances between clusters, revealing where the greatest variations in expression occur for a given channel in the compared clusters. These data are partially shown in Fig. 6. As per definition, not only FSC, SSC, DEAD-staining, Ly6G expression and CD11b expression were used by the algorithm to detect subpopulations, but also all other available information. However, since the other markers (CD19, CD3, CD8, CD4, F4/80) are generally not expressed on neutrophils, emphasis was much less from the algorithm on these markers, this is why we do not display them in Fig. 6. We added this information to the manuscript on page 7.

3. Along the same line, it would be important for future comparisons to include a table where all the identified subpopulations are described in the same manner as the above mentioned two.

Answer: We added a table with the files from the cluster explorer for all subpopulations, please see supplemental table 3.

---

## [Editor Report · Decision Letter 1]

18 Mar 2025

PONE-D-25-02396R1Using artificial intelligence-based software for an unbiased discrimination of immune cell subtypes in the fracture hematoma and bone marrow of non-osteoporotic and osteoporotic micePLOS ONE

Dear Dr. Haffner-Luntzer,

Thank you for submitting your manuscript to PLOS ONE. After careful consideration, we feel that it has merit but does not fully meet PLOS ONE’s publication criteria as it currently stands. Therefore, we invite you to submit a revised version of the manuscript that addresses the points raised during the review process.

**ACADEMIC EDITOR:**  Please submit print-quality figures. Kindly refer to the journals requirement for resolution and file size. It would help to re-check the quality after the PDF is generated to make sure the small font is still clear in print and on screen. Thank you!

We look forward to receiving your revised manuscript.

Kind regards,

Mohammed E. Elsalanty

Academic Editor

PLOS ONE

Journal Requirements:

Additional Editor Comments:

Authors have addressed all comments. One remark remains about the quality of the figures, which was poor and difficult to read especially in the resubmission. The authors are requested to submit high quality figures. Please test them in the collated PDF to make sure they do not lose quality during PDF generation.

---

## [Author Response · Author response to Decision Letter 2]

19 Mar 2025

We thank the editor for the important comment. We now provide the requested high resolution images. Please notice that in the build PDF, the images are still blurry, but when clicking on the provided download link, the images appeared in high resolution as submitted.

---

## [Editor Report · Decision Letter 2]

25 Mar 2025

Using artificial intelligence-based software for an unbiased discrimination of immune cell subtypes in the fracture hematoma and bone marrow of non-osteoporotic and osteoporotic mice

PONE-D-25-02396R2

Dear Dr. Haffner-Luntzer,

We’re pleased to inform you that your manuscript has been judged scientifically suitable for publication and will be formally accepted for publication once it meets all outstanding technical requirements.

Kind regards,

Mohammed E. Elsalanty

Academic Editor

PLOS ONE

Additional Editor Comments (optional):

Figures look good when downloaded individually but the quality on the collated PDF is still unreadable. Production team will need to make sure that the published PDF has adequate figure quality. No more concerns from editor.
---

## [Editor Report · Acceptance letter]

PONE-D-25-02396R2

PLOS ONE

Dear Dr. Haffner-Luntzer,

I'm pleased to inform you that your manuscript has been deemed suitable for publication in PLOS ONE. Congratulations! Your manuscript is now being handed over to our production team.

Kind regards,

on behalf of

Dr. Mohammed E. Elsalanty

Academic Editor

PLOS ONE